# Patterns and Predictors of Cervical Lymph Node Metastasis in Parathyroid Carcinoma

**DOI:** 10.3390/cancers14164004

**Published:** 2022-08-19

**Authors:** Ya Hu, Ming Cui, Xiaoyan Chang, Ou Wang, Tianqi Chen, Jinheng Xiao, Mengyi Wang, Surong Hua, Quan Liao

**Affiliations:** 1Department of General Surgery, State Key Laboratory of Complex Severe and Rare Disease, Peking Union Medical College Hospital, Chinese Academy of Medical Sciences & Peking Union Medical College, Beijing 100730, China; 2Department of Pathology, Peking Union Medical College Hospital, Chinese Academy of Medical Sciences & Peking Union Medical College, Beijing 100730, China; 3Department of Endocrinology, Key Laboratory of Endocrinology of the Ministry of Health, Peking Union Medical College Hospital, Chinese Academy of Medical Sciences & Peking Union Medical College, Beijing 100032, China; 4Department of Medical Research Center, Peking Union Medical College Hospital, Chinese Academy of Medical Sciences & Peking Union Medical College, Beijing 100730, China

**Keywords:** parathyroid carcinoma, lymph node metastasis, lymph node dissection, *CDC73*, survival

## Abstract

**Simple Summary:**

Parathyroid carcinoma (PC) is a rare endocrine malignancy with poor outcomes. Surgery remains the mainstay of PC treatment. However, due to the rarity of this malignancy, the optimal extent of PC surgery remains inconclusive, including whether to perform central lymph node dissection (LND). In the present study, we reported the patterns and predictors of cervical lymph node metastasis in PC based on a cohort of 68 PC patients. The results showed that the percentage of cervical lymph node metastasis in PC was 19.4% at initial surgery and 25.0% including reoperations for recurrencies. High-risk Schulte staging and *CDC73* abnormalities were identified as risk factors for cervical lymph node metastasis. Central LND should be considered during remedial surgeries performed after previous local resection of PC for patients with high risk factors.

**Abstract:**

Background: Parathyroid carcinoma (PC) is a rare endocrine malignancy with poor outcomes. Over 60% of PC patients experience repeated disease recurrence or metastasis. The significance of cervical lymph node dissection (LND) for PC remains inconclusive. Methods: PC patients diagnosed at Peking Union Medical College Hospital between 1992 and 2021 were reviewed retrospectively. Clinical data, initial tumor histological staging, parafibromin histochemical staining results, Ki67 index, *CDC73* gene mutation status and outcome information were collected systemically. The risk factors for recurrence and lymph node or distant metastasis were explored. Results: Sixty-eight PC patients receiving LND were enrolled. Cervical lymph node metastasis was identified in 19.4% of patients at initial surgery and 25.0% of patients including reoperations for recurrences. The independent risk factor for PC recurrence was a Ki67 index ≥ 5% (HR4.41, 95% confidence interval (CI)1.30–14.95, *p* = 0.017). Distant metastasis was an independent prognostic factor for PC patient overall survival (HR 5.44, 95% CI 1.66–17.82, *p* = 0.005). High-risk Schulte staging (*p* = 0.021) and *CDC73* abnormalities (*p* = 0.012) were risk factors for cervical lymph node metastasis. Conclusion: Most PCs were slow-growing, but lymph node metastasis was not rare. For patients planning to undergo remedial surgery after previous local resection of PC, central LND is suggested for tumors with high-risk Schulte staging or *CDC73* abnormalities.

## 1. Introduction

Primary hyperparathyroidism (pHPT) is the third most common endocrine disease. The proliferative parathyroid tissue autonomously secretes excessive parathyroid hormone (PTH), resulting in a series of clinical complications, such as hypercalcemia, urinary calculi, osteoporosis, bone fracture, and neuropsychiatric symptoms. Most primary hyperparathyroidism is caused by parathyroid adenoma or hyperplasia. Parathyroid carcinoma (PC) accounts for 0.5–1% of pHPT in western countries, while its incidence is approximately 5% in some countries, such as China and Japan [1,2,3,4,5]. The incidence of PC was reported to be significantly elevated in recent 20 years [6,7,8]. The outcome of PC is poor for over 60% of patients who experience local recurrence or metastasis. The leading cause of PC-related death is uncontrollable hypercalcemia and related complications rather than tumor invasion and destruction [2].

It is difficult to distinguish malignancies from benign parathyroid lesions before surgery. Several clues can indicate potential PC, such as serum calcium levels > 3.5 mmol/L, a 10-fold increase beyond the upper limit of the normal value of serum intact PTH (iPTH) and a palpable mass in the neck [9]. However, these clinical manifestations are not specific to PC as there is an obvious overlap of clinical presentations and biochemical parameters between malignant and benign parathyroid lesions. This problem is more prominent in developing countries where delayed diagnosis results in severe illness among patients with benign parathyroid lesions.

Identifying malignancy during surgery is also difficult [10]. Findings such as a firm and gray gland rather than soft and red tissue coated by a thick fibrous layer and adhesion with surrounding organs may indicate malignancy. However, these features are also not specific to malignancy. In addition, the accuracy of detecting PC with fast-frozen sections during surgery is only approximately 15% [11]. Surgery remains the mainstay of PC treatment as chemotherapy and radiotherapy are ineffective. Even though radical resection with en bloc resection of the parathyroid tumor and ipsilateral thyroid robe is suggested for PC in most published literature, the optimal excision extent is still in dispute [6,12,13,14]. The significance of ipsilateral cervical lymph node dissection (LND) during the initial surgery for PC has not been established. Most previous single-center studies indicate that radical resection helps to decrease the risk of recurrence, while the results obtained by analyzing large-scale cancer register databases, such as the SEER (Surveillance, Epidemiology, and End Result) database and the NCBD (National Cancer Database), did not confirm this finding [12,13,14,15]. This uncertainty regarding the extent of resection required interferes with PC management. Many PCs are misdiagnosed as benign lesions until the histological evaluation after initial resection. A dilemma for these patients is the necessity of radical remedial surgery. Several studies supported immediate remedial surgery, which was associated with better outcomes [1], while objections arose due to increased complications of reoperation and the lack of residual tumor identified in remedial surgery specimens [16,17]. Therefore, a more personalized protocol based on molecular classifications of PC may help to stratify patients likely to benefit from immediate remedial surgery.

Uncertainty regarding the significance of cervical LND during PC surgery is more prominent due to the scarcity of related clinical data. Although several studies have discussed the significance of LND for PC, no consensus has been reached [6,18,19,20]. According to the American Association of Endocrine Surgeons guidelines, prophylactic central or lateral LND is not routinely suggested for PC patients, even though the evidence against the procedure is insufficient [21]. In addition, most previous studies focused on the general analysis of data derived from cancer registry centers or small cohorts, in which the details of lymph node metastasis over the course of PC were lacking.

In the present study, the outcomes of a cohort of 68 PC patients were analyzed. The risk factors for lymph node metastasis in PC were investigated, including molecular biomarkers, such as *CDC73* abnormalities and the Ki67 index. The primary purpose of this retrospective study was to identify the patterns and predictors of cervical lymph node metastasis in PC. Risk factors associated with the survival of PC patients were also investigated.

## 2. Materials and Methods

### 2.1. Subjects

Patients diagnosed with PC at Peking Union Medical College Hospital (PUMCH), a tertiary teaching hospital, were enrolled retrospectively in this study between 1992 and 2021. Cases without reliable information were excluded. All available clinical information was reviewed from medical records from a previous institution or our own. Outcome data, such as disease-free survival (DFS) and overall survival (OS) times, were collected from outpatient service records or during follow-up. The normal ranges of serum calcium level and serum iPTH level in our hospital are 2.13–2.70 mmol/L and 12–65 pg/mL, respectively. En bloc surgery was defined as resection of at least the parathyroid lesion and hemithyroidectomy with or without ipsilateral central LND. Simple resection of parathyroid lesions was considered local resection. PC diagnosis was based on histopathological examination according to the 2017 WHO criteria [22]. PC was classified as high or low risk of recurrence based on a staging system developed by Talat and Schulte in 2010 [20]. PC with high-risk staging was identified by histological evidence of cancer invasion into blood vessels or vital organs, lymph node metastasis or distant metastasis during the initial surgery [23]. Cystic PC was identified by preoperative neck ultrasound. Biochemical remission was defined when serum calcium levels dropped and remained below the upper limit of the normal reference range for over six months after surgery. Recurrence was defined when serum calcium levels exceeded the upper normal limit after the patient had achieved remission. The study was approved by the Ethics Committee of Peking Union Medical College Hospital (S-K1743), and written informed consent was obtained from the participants.

### 2.2. Immunohistochemical Staining and CDC73 Mutation Status

Most of the parafibromin and Ki67 immunohistochemical (IHC) staining results and *CDC73* mutation status data were collected from previous studies at our institute [24,25,26,27]. Some of the IHC results and *CDC73* sequencing were supplemented according to our previous studies mentioned above. In brief, immunohistochemical staining for parafibromin was performed on FFPE tumor samples. Tissue sections were deparaffinized and dehydrated and then incubated in a 3% H_2_O_2_ solution. Antigen-retrieval solution containing ethylenediaminetetraacetic acid was used for heat-induced antigenic retrieval. A total of 3% bovine serum albumin was used to block nonspecific antibody binding sites, and primary antibodies against parafibromin (ab223840, Abcam, Cambridge, UK) were used to incubate the slides overnight at 4 °C. Secondary antibodies and diaminobenzidine were used for staining. Genomic DNA was extracted from frozen tumor tissues or formalin-fixed, paraffin-embedded tumor samples. DNA concentration and quality was evaluated. Genomic DNA samples were fragmented by sonication. The sequencing library was generated and sequenced on the Illumina HiSeq platform. Clean reads were mapped to the reference human genome (GRCh37). Single nucleotide variations (SNVs) and small insertions and deletions (indels) were called with MuTect or GATK. Negative parafibromin IHC staining was defined as staining loss in all nuclei of tumor cells, with vascular endothelial cells and stromal cells as internal positive controls. *CDC73* mutation was identified as somatic or germline mutation identified by second-generation DNA sequencing of cancer tissues. For the cases with inconsistent results between parafibromin staining and DNA sequencing, the former was used for our analysis. Parafibromin staining loss or *CDC73* mutations detected by DNA sequencing were uniformly defined *CDC73* abnormalities [5,28].

### 2.3. Statistical Analysis

Continuous variables are presented as the mean value ± standard deviation (SD). Discrete data were described as case numbers or corresponding percentages. Survival times were calculated from the date of the initial surgery to the date of death, first relapse or last contact. Fisher’s exact test was used to compare categorical variables, and *t* test was used to compare continuous variables. The Kaplan–Meier method was used to perform survival analyses. A Cox proportional hazards regression analysis was applied to evaluate the effects of variables on DFS and OS outcomes with hazard ratios (HRs). Factors with a *p* value < 0.05 in the univariate analysis were used for the multivariate analysis. When datasets regarding special variables were incomplete, the exact case numbers used in this study were indicated. SPSS Statistics for Windows version 16 (SPSS Inc., Chicago, IL, USA) and GraphPad Prism 6 (GraphPad Software, Inc., San Diego, CA, USA) were used for the statistical analysis. *p* values < 0.05 with 2-tailed tests were considered statistically significant.

## 3. Results

### 3.1. Clinical Characteristics of Patients with PC

The prevalence of PC in all pHPT patients that underwent surgery at PUMCH is approximately 3.9%. Out of the 105 patients diagnosed with PC, 68 with lymph node status data were included in the study (Table 1). The male to female ratio was 1:0.94, and the average age at primary diagnosis was 42.5 years (range 17–71 years). The mean serum calcium levels were 3.57 ± 0.62 mmol/L, ranging from 2.33 to 4.85 mmol/L. The mean serum iPTH levels were 1319.3 ± 776.4 pg/mL, ranging from 46.9 to 3875.0 pg/mL. Hypercalcemic crisis with severe clinical symptoms was observed in 22.1% of patients. Cystic PC accounted for 18.3% of patients. Two patients had a history of hereditary hyperparathyroidism, including a *CDC73* germline mutation in one patient and an unknown cause in the other. External radiation therapy was performed on four patients as adjuvant therapy after the initial surgery. Chemotherapy regimen TPF (docetaxel, cisplatin and fluorouracil) was applied in one patient. Molecular targeted therapy such as anlotinib, sorafenib, apatinib, and gefitinib were administered to four patients. Parafibromin staining loss or *CDC73* mutations were identified in 29 (53.7%) patients among 54 patients with available data (Figure 1). The Ki67 index of primary tumor specimens was available for 52 patients, which was ≥5% in 36 patients (69.2%) and ≥10% in 23 (44.2%).

A total of 187 surgeries were performed on 68 PC patients (median, 2; range, 1–12), including 175 cervical operations. The first surgery was local resection in 26 patients and en bloc resection with or without lymphadenectomy in 35 and seven patients, respectively. Three patients experienced distant metastasis before the initial surgery including two patients with lung metastasis and one patient with bone metastasis. Six patients did not achieve biochemical remission after the initial surgery, including one patient (case 34) who presented re-elevation of serum calcium levels and iPTH levels and was identified with lung metastasis in the fourth month postoperatively. Thus, the above nine patients were excluded from the DFS outcome calculation. A total of 41 patients experienced recurrence, including distant metastasis in 14 patients. The distant metastatic sites included lung metastasis (*n* = 11), bone metastasis (n = 1), combined lung/bone metastasis (*n* = 1) and combined lung/brain metastasis (*n* = 1). The DFS rates of our cohort were 33.3% at 5 years and 16.0% at 10 years (Figure 2). The median interval between the initial operation and recurrence was 2.6 years (range 0.5–17 years). 

Among the 68 patients, 17 (25%) died, and the median follow-up time was 5.1 years (0.5–26.9 years). The causes of death included hypercalcemia in 15 patients and other comorbidities in one patient; the cause of death was unknown in one patient. The 5-year and 10-year overall survival rates were 86.8% and 71.4%, respectively (Figure 2).

### 3.2. Factors Influencing DFS and OS Rates of PC Patients

In the univariate Cox regression, the factors associated with recurrence were high calcium levels, hypercalcemic crisis, *CDC73* abnormalities and a Ki67 index ≥ 5% (Table 2). In the multivariate Cox regression, a Ki67 index ≥ 5% (HR 4.41, 95% CI 1.30–14.95, *p* = 0.017) was the only independent risk factor associated with recurrence. Compared with local resection, en bloc surgery as the initial surgery was not related to DFS (*p* = 0.707) or OS (*p* = 0.060) outcomes.

In the univariate Cox analysis, high-risk Schulte staging and distant metastasis were identified as prognostic factors for OS outcomes (Table 2). In the multivariate Cox regression analysis, distant metastasis (HR 5.44, 95% CI 1.66–17.82, *p* = 0.005) was the only factor independently associated with OS outcomes.

### 3.3. Predictive Factors for Lymph Node Metastasis

Detailed information on lymph node metastasis status was available for 68 patients. For the initial surgery, 35 patients underwent central LND and one patient (case 50) underwent lateral LND. Lymph node metastasis was identified in seven (19.4%) patients. Among the initial and subsequent surgeries, 17 (25%) patients had metastasis in a total of 29 positive lymph nodes (median, 1; range, 1–6) (Figure 3). Central LND was performed on 67 patients, 35 during the initial surgery and 32 during remedial surgery for recurrence, and a total of 17 metastatic lymph nodes were identified in 10 of these patients (median, 1; range, 1–6). Lateral LND was performed on 17 patients, one during the initial surgery and 16 during remedial surgery for recurrence, and a total of 12 metastatic lymph nodes were identified in eight of these patients (median, 1.5; range, 1–2). 

The risk factors for lymph node metastasis in PC were high-risk Schulte staging (*p* = 0.021) and *CDC73* abnormalities (*p* = 0.012). Other factors, such as sex (*p* = 0.580), age (*p* = 0.345), diameter ≥ 3 cm (*p* = 0.554), a palpable mass in the neck (*p* = 0.081), serum iPTH (*p* = 0.541), serum calcium level (*p* = 0.079), hypercalcemic crisis (*p* = 0.177), or a Ki67 index ≥ 5% (*p* = 0.730), were not found to be related to lymph node metastasis risk.

Postoperative recurrent laryngeal nerve (RLN) paralysis occurred in three patients after the first surgery, including in one patient in whom resection of the RLN was intentionally performed to avoid residual tumor. The rate of accidental RLN injury during the initial surgery was similar between local resection and en bloc resection approaches (2 vs. 1, *p* = 0.553).

## 4. Discussion

Due to the low incidence of PC, several key problems regarding the clinical management of this malignancy remain unresolved, such as preoperative diagnosis approaches and determination of the optimal extent of resection. A considerable proportion of PC patients are diagnosed after postoperative histopathological examination of locally resected tissue obtained during the initial surgery. The necessity of remedial surgery has not yet been determined, and subsequent radical resection poses significantly increased risk for morbidities, such as recurrent laryngeal nerve palsy. Furthermore, it has not yet been elucidated whether it is necessary to perform cervical LND for PC patients. Based on a relatively large cohort, the present study might provide new clues to assist clinical decision-making.

The optimal extent of resection for PC has long been debated, and most studies recommend en bloc resection for PC based on oncological principles [4,9,29,30]. Improved outcomes with en bloc resection have been revealed by some single-center retrospective studies or case reviews, while this survival advantage could not be validated by analyzing cancer registries with larger sample sizes. Publishing bias was unavoidable by principle as a positive result might be more likely to be published. In our study, the extent of resection at the initial operation was not found to be associated with the risk for recurrence or death. This result concurs with those of previous studies of tumor registry data, such as the SEER database or NCDB [6,12,14]. One possible reason for this finding is that the prominent invasive pattern of PC found during surgery might prompt the surgeon to perform radical surgery [31]. This may also be a possible reason why the overall survival rate of patients who received en bloc resection tended to be worse in our study, although the difference was not statistically significant. For tumors suspected to be malignant before or during surgery, we still recommend en bloc resection considering the risk of complications and difficulties associated with reoperation.

In our present study, high serum calcium levels before surgery, *CDC73* abnormalities, and a Ki67 index ≥ 5% were associated with PC recurrence. High-risk Schulte staging and distant metastasis were unfavorable prognostic factors, while distant metastasis was the only independent factor related to OS outcomes. Our previous meta-analysis showed that parafibromin staining loss was associated with PC recurrence [32]. Parafibromin is encoded by *CDC73*, a tumor suppressor gene, and inactivating mutations in *CDC73* cause hyperparathyroidism-jaw tumor syndrome. *CDC73* mutation was first identified in PC by Howell and has subsequently been found in around 60% of PC patients [33,34,35,36,37,38,39]. Studies showed that parafibromin staining loss rather than *CDC73* mutations was associated with worse overall survival [28,32]. The Ki67 index detected by immunohistochemistry is the most commonly used marker to evaluate tumor cell proliferation. Our results showed that a higher Ki67 index was associated with recurrence, consistent with the results of previous studies [30,40]. Instead of a refined TNM classifications dependent on tumor and lymph node status, the Schulte staging system that simply classifies by high or low risk was applied in our present study for its simplicity and clinical practicability [20]. We found that high-risk Schulte staging of PC strongly indicated an adverse prognosis, consistent with the results of previous studies [1,20]. Therefore, en bloc surgery may be more beneficial for PC patients with *CDC73* abnormalities, a tumor tissue Ki67 index ≥ 5%, and high-risk Schulte staging. However, the diagnosis of PC as well as relevant molecular information may only be achieved after initial surgery for the majority of patients.

The lymph node metastasis rate was reported to range from 0 to 42.8% in previous studies [6,41]. In our cohort, the lymph node metastasis rate was 19.4% in the patients who underwent cervical LND during the initial surgery and 25% in the patients who underwent LND overall. Several studies have revealed that lymph node metastasis in PC is associated with worse prognoses [20,30,31]. Although the difference was not statistically significant, it was found that LND tended to decrease the recurrence risk in our study. However, due to the retrospective design of the present study, it is not sufficient to elucidate the significance of cervical LND during surgery for PC. More prospective randomized clinical trials are needed to address this issue.

More importantly, we found that *CDC73* abnormalities and high-risk Schulte staging were associated with lymph node metastasis. It is difficult to detect the *CDC73* status of parathyroid neoplasms before surgery, as fine needle aspiration of parathyroid tumors was not recommended to avoid tumor rupture, seeding and bleeding [21,42]. However, for PC patients receiving local resection as the initial operation, IHC staining of the resected tumor samples could provide these critical indexes [43]. Therefore, central LND is recommended during remedial surgery for PC patients with *CDC73* abnormalities and high-risk Schulte staging. Lateral LND might not be necessary without definitive imaging evidence of metastasis.

Several limitations of this study should be noted. First, due to the low incidence of PC, even though the present study may represent the largest single-center cohort analyzed, the limited sample size might still affect the evaluation of the association between factors and patient prognoses. Second, data regarding preoperative PTH values, parafibromin staining results or Ki67 index values were unavailable for some patients, who were misdiagnosed with thyroid tumors until postoperative histopathological examination or who received the initial surgery at a different hospital in the past. The number of cases available for analysis for each specific feature was indicated. Collectively, the retrospective design of the present study may have introduced bias. Prospective studies with larger cohorts are warranted to validate our findings.

## 5. Conclusions

In conclusion, although PC is a slow-growing malignancy throughout the overall course of disease, lymph node metastasis was not a rare event. High-risk Schulte staging and *CDC73* abnormalities could predict lymph node metastasis risk in PC patients. For patients planning to undergo remedial surgery after previous local resection of PC, central LND is suggested for tumors with high-risk Schulte staging or *CDC73* abnormalities.

## Figures and Tables

**Figure 1 cancers-14-04004-f001:**
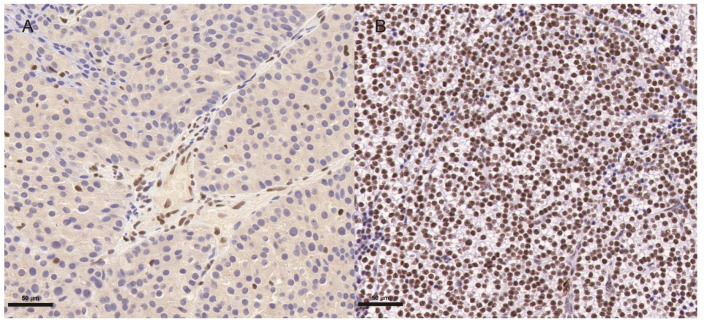
Representative images of parafibromin staining in parathyroid carcinoma tissue. (**A**) Parafibromin staining loss (Case 43). (**B**) Positive parafibromin staining (Case 7). Scale bar, 50 μm.

**Figure 2 cancers-14-04004-f002:**
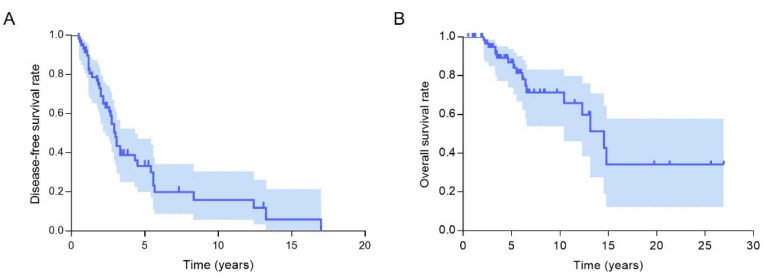
Cumulative disease-free survival (**A**) and overall survival (**B**) curves for patients with parathyroid carcinoma.

**Figure 3 cancers-14-04004-f003:**
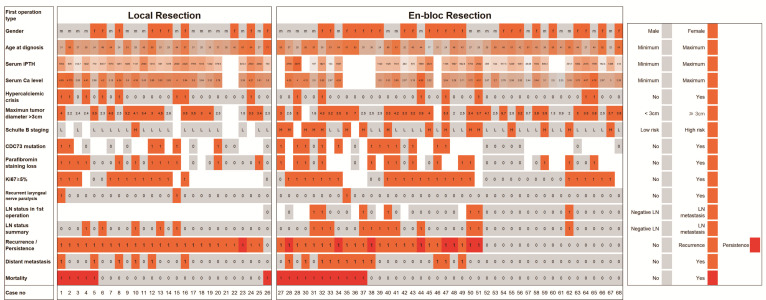
The clinical characteristics, *CDC73* abnormalities, Ki67 index, and lymph metastasis status of 68 patients with PC.

**Table 1 cancers-14-04004-t001:** Clinical characteristics of patients with parathyroid carcinoma (*n* = 68 unless otherwise specified).

Characteristics	Value
Average age at diagnosis (years)	42.5 ± 13.09
Male:Female	35:33
Initial surgery performed at our institute	39.7%
Serum calcium level before initial operation (mmol/L) (*n* = 60)	3.57 ± 0.62
Serum intact parathyroid hormone (pg/mL) level before initial operation (*n* = 57)	1319.3 ± 776.4
Hypercalcemic crisis (%)	22.1%
Maximum diameter of initial focus (cm) (*n* = 63)	3.25 ± 1.16
Cystic tumors (%) (*n* =60)	18.3%
*CDC73* abnormalities (%) (*n* = 54) Parafibromin staining loss (%) (*n* = 50) *CDC73* mutations (%) (*n* = 35)	53.7% 52% 51.4%
Ki67 index ≥ 5% of initial tumor specimens (%) (*n* = 52)	69.2%
Patients with a family history of hyperparathyroidism	2

**Table 2 cancers-14-04004-t002:** Risk factors for the disease-free survival and overall survival outcomes of PC patients in the univariate Cox proportional hazard analysis.

Factors	Disease-Free Survival			Overall Survival		
	Hazard Ratio	Confidence Interval	*p* Value	Hazard Ratio	Confidence Interval	*p* Value
Age at diagnosis (years)	1.00	0.98–1.02	0.997	1.03	0.99–1.06	0.152
Male vs. Female	1.06	0.57–1.98	0.851	1.24	0.47–3.29	0.665
Cystic tumors (%) (*n* = 60)	0.70	0.29–1.72	0.441	0.52	0.07–4.03	0.527
Maximum diameter of initial tumor (*n* = 63)	1.06	0.79–1.42	0.689	0.87	0.57–1.32	0.520
Palpable mass in the neck (*n* = 63)	0.98	0.50–1.91	0.946	0.60	0.21–1.70	0.335
Serum iPTH at diagnosis (pg/mL) (*n* = 57)	1.00	1.00–1.00	0.286	1.00	1.00–1.00	0.821
Serum calcium level at diagnosis (*n* = 60)	2.90	1.50–5.59	0.001 **	1.59	0.58–4.38	0.367
Hypercalcemic crisis	2.44	1.17–5.11	0.018 *	1.53	0.53–4.37	0.431
Extent of dissection (En bloc resection vs. local resection)	0.88	0.45–1.71	0.707	2.66	0.96–7.36	0.060
Schulte staging (high risk vs. low risk) (*n* = 60)	1.79	0.80–3.97	0.155	4.84	1.52–15.35	0.007 **
Lymph node dissection in first surgery (*n* = 36)	0.40	0.15–1.08	0.072	2.89	0.54–15.63	0. 217
*CDC73* abnormalities (*n* = 54)	3.53	1.64–7.63	0.001 **	2.73	0. 81–9.23	0.106
Ki67 index ≥ 5% (*n* = 52)	4.52	1.56–13.10	0.005 **	1.08	0.29–4.03	0.906
Number of cumulative operations				1.03	0.85–1.25	0.784
Lymph node metastasis				1.77	0.59–5.30	0.311
Distant metastasis				3.80	1.43–10.06	0.007 **

* *p* < 0.05, ** *p* < 0.01.

## Data Availability

All related clinical data in this manuscript are stored and can be accessed from our clinical database of parathyroid carcinoma.

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
