# Peer review of "Patterns and Predictors of Cervical Lymph Node Metastasis in Parathyroid Carcinoma"

_cancers, 2022, doi:10.3390/cancers14164004_

Round 1
Reviewer 1 Report
Accept in present form
Reviewer 2 Report
The authors addressed all the comments from reviewers by modifying the manuscript. I recommend the manuscript for publication.
Reviewer 3 Report
no further comments
This manuscript is a resubmission of an earlier submission. The following is a list of the peer review reports and author responses from that submission.
Round 1
Reviewer 1 Report
Hu et al. evaluated the outcomes of a cohort di 68 patients with PC particularly risk factors of lymph node metastasis. The authors found that the percentage of lymph node metastases was 25%. Moreover, they found that high-risk Schulte staging and CDC73 mutations were identified as risk factors for cervical lymph node metastasis. The authors concluded that prophylactic central compartment LND should be considered during initial surgery. Moreover, for patients planning remedial surgery after previous local resection of PC, central LND is suggested for tumors with high-risk Schulte staging or CDC73 mutations.
Major comments
The paper is well written and interesting. The data could be useful in the field.
However, some comments and suggestions should be addressed.
Lines 104, the authors should report that all but 2 sporadic primary hyperparathyroidism. The authors report in table 1 that 2 patients had family history of primary hyperparathyroidism. Could you give some details regarding on what familial form we are dealing with?
Lines 53-54 Other authors in Italy found a prevalence of PC around 0.5%. This prevalence should also be added in the text and the reference should be quoted (European Journal of Endocrinology, 2007, 156: 547-554).
Lines 54-55 The incidence of PC has also reported by other authors and different countries and should be added in the text and references (ANZ J Surg 2011: 81: 528-532; Acta Oncol 2017: 56: 991-1003)
Line 74, the authors stated that “en bloc resection… “ references are missing
Line 84, the authors stated that “several studies supported immediate remedial ….” references are missing
Lines 109-110, the authors state that “en bloc resection… with or without ipsilateral central LND”. The authors should define on which criteria was chosen to perform local excision rather than en bloc resection. Moreover, how did they decide to perform excision of lymph nodes or not? On what criteria? Did they evidence of suspected enlarged lymph nodes at neck imaging?
Lines 111, the authors state that “resection of parathyroid lesion with or without partial thyroidectomy was considered local resection”. The authors should specify whether the partial thyroidectomy was performed for thyroid disease or because the Pc was infiltrating the thyroid of for other reasons. I would indicate this group of patients as “simple PTX” if the partial thyroidectomy was carried out for thyroid disease.
Lines 118, the authors should better define the concept of recurrence. Do they mean after at least 6 months of normocalcemia?
2.2 paragraph, the authors should separate immunohistochemistry from CDC73 genetic analysis. The authors should report the protocol for CDC73 analysis and should also specify whether beside point mutations, gross deletions/insertions were evaluated.
The authors should also report the protocol for immunohistochemistry.
For better clarity for the reader, the results in table 1 and Figure 2 should also be reported separately
Lines 132, I agree with the authors that patients with negative parafibromin typically had CDC73 mutations, but this is not true in all patients. Thus, I would separate the immune data from those of genetic analysis and report separately
Lines 156, it would be of interest to report whether external radiation was done as adjuvant therapy or for other reasons.
Lines 157, the authors state that systemic chemotherapy or molecular targeted therapy administered to 5 patients. The authors should specify the clinical and genetic profile of these 5 pts and the specific treatment they received.
Lines 164, please give details regarding distant metastasis
Lines 167, the authors state that a total of 46 pts had recurrence or metastasis, including cervical recurrence in 42 and distant metastasis in 18 pts (in total 60). Thus, it seems that some patients had either local recurrence or distant metastases. Please better specify. Moreover, it should be added the sites of metastases.
Lines 170-172, it would be clearer state how many surgeries (mean/median) had each patient instead of reporting the total number of surgeries. Moreover, it should be specify whether the decision to perform lymph node dissection was based on the evidence of enlarged lymph nodes at neck imaging (neck ultrasound, CT or ..)
Lines 195, predictive factors for lymph node metastasis. This paragraph should be re-written because it is confusing for the reader. If I understand well, 646 LNs were performed but only 32 were metastasis. I would specify how many patients had lymph node metastasis (mean/median node dissection), and it seems only 17/68 (25%).
Discussion, lines 228-229, references are missing
Lines 249-251. The information reported is not corrected. The first author that reported CDC73 mutation in PC was Howell et al (2003), followed in the same year by Shattuck et al (2003) and then Cetani et al (2004). The authors should revise the references on the CDC73 mutations in PC and immunohistochemical data in PC.
The authors should better add in the discussion the notion that parafibromin loss but not CDC73 mutations is associated with a lower overall survival (also see: CDC73 mutational status and loss of parafibromin in the outcome of parathyroid cancer. Endocr Connections 2013; Downregulation of CASR expression and global loss of parafibromin staining are strong negative determinants of
prognosis in parathyroid carcinoma. Modern Pathology 2011 24 688–697).
Lines 258-259, the sentence the authors report is correct, however in most patients the diagnosis of PC is done after surgery except in those who have metastasis pre-first surgery. I would add a sentence dealing with this issue.
A comment on the last WHO criteria should be added in the discussion (Overview of 2022 WHO Classification of Parathyroid Tumors, Endocrine Pathology 2022)
Table 1. The authors should better explain how they define “cystic tumors”. Was this feature based on ultrasound pattern or histological finding?
Figure 1, since the results in the text are reported in “years”, I would also change the ordinate axis as “years” and not “months”
Author Response
Dear reviewer,
Thank you for your professional and valuable comments. We have revised our manuscript based on these comments, and we hope that these revisions will be met with approval. The changes are highlighted in the revised manuscript, and our point-by-point responses are attached below.
Point 1: Lines 104, the authors should report that all but 2 sporadic primary hyperparathyroidism. The authors report in table 1 that 2 patients had family history of primary hyperparathyroidism. Could you give some details regarding on what familial form we are dealing with?
Thank you for the comment. For the first patient, her father, brother, daughter, and aunt were diagnosed as primary hyperparathyroidism. For the second patient, his two sisters, daughter, nephew, and niece were diagnosed as primary hyperparathyroidism and underwent surgeries. The pathology indicated parathyroid carcinoma in one of his sisters and parathyroid adenomas in other relatives. Regrettably, no further sequencing was performed for the second kindred.
Point 2: Lines 53-54 Other authors in Italy found a prevalence of PC around 0.5%. This prevalence should also be added in the text and the reference should be quoted (European Journal of Endocrinology, 2007, 156: 547-554).
Thank you for the comment. We have revised the description in the manuscript and added the reference.
Point 3: Lines 54-55 The incidence of PC has also reported by other authors and different countries and should be added in the text and references (ANZ J Surg 2011: 81: 528-532; Acta Oncol 2017: 56: 991-1003)
Thank you for the comment. We have added above two references into the manuscript.
Point 4: Line 74, the authors stated that “en bloc resection… “ references are missing
Thank you for the comment. We have added relevant references.
Point 5: Line 84, the authors stated that “several studies supported immediate remedial ….” references are missing
Thank you for the comment. We have added relevant references.
Point 6: Lines 109-110, the authors state that “en bloc resection… with or without ipsilateral central LND”. The authors should define on which criteria was chosen to perform local excision rather than en bloc resection. Moreover, how did they decide to perform excision of lymph nodes or not? On what criteria? Did they evidence of suspected enlarged lymph nodes at neck imaging?
Thank you for the comment. The present study is a retrospective study including a portion of patients who underwent initial surgery in other institutions and were referred to our hospital for further management. Thus, it is hard to define the criteria for all enrolled patients. In our institute, en bloc resection accompanied with routinely ipsilateral cervical LND is increasingly performed as standard surgery for PC at present. This procedure will be performed if there is any presence of peripheral invasion, enlarged lymph node or suspicion of malignancy indicated by intraoperative frozen sections.
Point 7: Lines 111, the authors state that “resection of parathyroid lesion with or without partial thyroidectomy was considered local resection”. The authors should specify whether the partial thyroidectomy was performed for thyroid disease or because the Pc was infiltrating the thyroid of for other reasons. I would indicate this group of patients as “simple PTX” if the partial thyroidectomy was carried out for thyroid disease.
Thank you for the comment. Our statement should be revised. The main reason for partial thyroidectomy in our cohort is to expand the surgical extent for parathyroid tumors. We have modified the statement in the manuscript accordingly.
Point 8: Lines 118, the authors should better define the concept of recurrence. Do they mean after at least 6 months of normocalcemia?
Thank you for the comment. Remission was defined when serum calcium dropped and remained below the upper limit of the normal reference range for over 3 months after surgery. Recurrence was defined when serum calcium exceeded the upper normal limit after the patient had achieved remission. We have modified the statement in the manuscript accordingly.
Point 9: 2.2 paragraph, the authors should separate immunohistochemistry from CDC73 genetic analysis. The authors should report the protocol for CDC73 analysis and should also specify whether beside point mutations, gross deletions/insertions were evaluated. The authors should also report the protocol for immunohistochemistry. For better clarity for the reader, the results in table 1 and Figure 2 should also be reported separately
Thank you for the comment. We have separated the results of CDC73 analysis into IHC part and sequencing part in Table 1 and Figure 2. We have also added the protocol of CDC73 analysis in the method part (Lines 127-139).
Point 10: Lines 132, I agree with the authors that patients with negative parafibromin typically had CDC73 mutations, but this is not true in all patients. Thus, I would separate the immune data from those of genetic analysis and report separately
Thank you for the comment. We have separated the IHC data and sequencing data. The definition of CDC73 abnormalities was used to conclude parafibromin loss and CDC73 mutations according to previous literatures (Should parafibromin staining replace HRTP2 gene analysis as an additional tool for histologic diagnosis of par-athyroid carcinoma? Eur. J. Endocrinol. 2007, 156, 547-554. CDC73 mutational status and loss of parafibromin in the outcome of parathyroid cancer. Endocr Connect 2013, 2, 186-195.)
Point 11: Lines 156, it would be of interest to report whether external radiation was done as adjuvant therapy or for other reasons.
Thank you for the comment. External radiation was done as adjuvant therapy. We have modified the statement in the manuscript.
Point 12: Lines 157, the authors state that systemic chemotherapy or molecular targeted therapy administered to 5 patients. The authors should specify the clinical and genetic profile of these 5 pts and the specific treatment they received.
Thank you for the comment. Anlotinib was applied for two patients. Gifitinib was applied for one patient carrying the EGFR L858R mutation. Sorafenib and sequential apatinib was applied for one patient. Chemotherapy regimen TPF (docetaxel, cisplatin and fluorouracil) was applied for one patient. Unfortunately, above systemic treatment showed no notable effect on these patients until now. We have revised the statement in the manuscript.
Point 13: Lines 164, please give details regarding distant metastasis
Thank you for the comment. Three patients experienced distant metastases before the first surgery including two patients with lung metastasis and one patient with bone metastasis. We have modified the statement in the manuscript.
Point 14: Lines 167, the authors state that a total of 46 pts had recurrence or metastasis, including cervical recurrence in 42 and distant metastasis in 18 pts (in total 60). Thus, it seems that some patients had either local recurrence or distant metastases. Please better specify. Moreover, it should be added the sites of metastases.
Thank you for the comment. We agree that our current statements on recurrence status of patients were indeed formulated unclearly. We originally classified patients with new-onset cervical lesions who had persistent disease into the cervical recurrence group, however, this may arise unnecessary confusion. Thus, we deleted the description of cervical recurrence and revised the relevant statement in the manuscript. In addition, there were 15 patients experience distant metastasis after initial surgery (excluding 3 patients experiencing distant metastasis before the initial surgery). The distant metastatic sites included lung metastasis (n = 12), bone metastasis (n = 1), combined lung/bone metastasis (n = 1), and combined lung/brain metastasis (n = 1). We have revised the statement in the manuscript.
Point 15: Lines 170-172, it would be clearer state how many surgeries (mean/median) had each patient instead of reporting the total number of surgeries. Moreover, it should be specify whether the decision to perform lymph node dissection was based on the evidence of enlarged lymph nodes at neck imaging (neck ultrasound, CT or ..)
Thank you for the comment. The median number of surgeries was 2 (range, 1-12). We have revised the statement in the manuscript. Since the initial surgeries were performed in different institutions, the decision criteria was not available for all patients. According to our opinion in recent years, lymph nodes dissection is a routine procedure for patients receiving en bloc resection and remedial surgery for recurrence if it is possible.
Point 16: Lines 195, predictive factors for lymph node metastasis. This paragraph should be re-written because it is confusing for the reader. If I understand well, 646 LNs were performed but only 32 were metastasis. I would specify how many patients had lymph node metastasis (mean/median node dissection), and it seems only 17/68 (25%).
Thank you for the comment. We have modified the statement in the manuscript.
Point 17: Discussion, lines 228-229, references are missing
Thank you for the comment. We have added relevant references.
Point 18: Lines 249-251. The information reported is not corrected. The first author that reported CDC73 mutation in PC was Howell et al (2003), followed in the same year by Shattuck et al (2003) and then Cetani et al (2004). The authors should revise the references on the CDC73 mutations in PC and immunohistochemical data in PC.
Thank you for the comment. We have revised the statement in the manuscript.
Point 19: The authors should better add in the discussion the notion that parafibromin loss but not CDC73 mutations is associated with a lower overall survival (also see: CDC73 mutational status and loss of parafibromin in the outcome of parathyroid cancer. Endocr Connections 2013; Downregulation of CASR expression and global loss of parafibromin staining are strong negative determinants of prognosis in parathyroid carcinoma. Modern Pathology 2011 24 688–697).
Thank you for the comment. We have added the statement and relevant references in the manuscript (Lines 270-273).
Point 20: Lines 258-259, the sentence the authors report is correct, however in most patients the diagnosis of PC is done after surgery except in those who have metastasis pre-first surgery. I would add a sentence dealing with this issue.
Thank you for the comment. We have added the following sentence in the revised manuscript: “However, the diagnosis of PC as well as relevant molecular information may only be achieved after initial surgery for the majority of patients.”
Point 21: A comment on the last WHO criteria should be added in the discussion (Overview of 2022 WHO Classification of Parathyroid Tumors, Endocrine Pathology 2022)
Thank you for the comment. We have added the reference in the revised manuscript (Line 301).
Point 22: Table 1. The authors should better explain how they define “cystic tumors”. Was this feature based on ultrasound pattern or histological finding?
Thank you for the comment. Cystic tumors were identified by neck ultrasound. We have added relevant statement in the revised manuscript (Lines 116-117).
Point 23: Figure 1, since the results in the text are reported in “years”, I would also change the ordinate axis as “years” and not “months”
Thank you for the comment. We have revised Figure 1 accordingly.
Reviewer 2 Report
In the manuscript titled as ‘Patterns and Predictors of Cervical Lymph Node Metastasis in Parathyroid Carcinoma’, the authors attempt to identify the predicting factors for cervical lymph node metastasis in Parathyroid carcinoma. They demonstrate that high-risk schulte staging and CDC73 mutations were the risk factors for cervical lymph node metastasis in parathyroid carcinoma.
The manuscript is well written, scientifically sound and the findings generated in this study is clinically relevant. However, I have a few concerns regarding the presentation of the results. For instance, the title of the manuscript (Patterns and Predictors of Cervical Lymph Node Metastasis in Parathyroid Carcinoma) does not directly describe the findings of the study. The title does not specify the predictors of cervical lymph node metastasis in parathyroid carcinoma. It would be better if the authors stated in the title which factors predict cervical lymph metastases.
As discussed by the authors, the low sample size is a limitation of this study. If authors could discuss recent papers with large sample sizes regarding predictors of parathyroid carcinoma in the discussion section, that would be great. For instance, the authors did not discuss an important work (Prognostic Analysis for Patients with Parathyroid Carcinoma: A Population-Based Study, PMID: 35250443) that describes the predictive factors of parathyroid carcinoma. Please include such updated information while discussing about the predictive factors of parathyroid carcinoma.
The authors could consider presenting the representative IHC figures of parafibromin in patient samples in results section because the results shows that the mutations in the genes (CDC73 mutations) that encodes the parafibromin is a predictive factor for cervical lymph node metastasis in parathyroid carcinoma.
Author Response
Dear reviewer,
Thank you for your professional and valuable comments. We have revised our manuscript based on these comments, and we hope that these revisions will be met with approval. The changes are highlighted in the revised manuscript, and our point-by-point responses are attached below.
Point 1: the title of the manuscript (Patterns and Predictors of Cervical Lymph Node Metastasis in Parathyroid Carcinoma) does not directly describe the findings of the study. The title does not specify the predictors of cervical lymph node metastasis in parathyroid carcinoma. It would be better if the authors stated in the title which factors predict cervical lymph metastases.
We thank the reviewer for this comment. While it is true that the predictive factors for cervical lymph node metastasis of PC are main findings in our manuscript, they may not conclude all results. We respectfully request to keep the title.
Point 2: As discussed by the authors, the low sample size is a limitation of this study. If authors could discuss recent papers with large sample sizes regarding predictors of parathyroid carcinoma in the discussion section, that would be great. For instance, the authors did not discuss an important work (Prognostic Analysis for Patients with Parathyroid Carcinoma: A Population-Based Study, PMID: 35250443) that describes the predictive factors of parathyroid carcinoma. Please include such updated information while discussing about the predictive factors of parathyroid carcinoma.
Thank you for the comment. We have added the reference in the revised manuscript (Line 80).
Point 3: The authors could consider presenting the representative IHC figures of parafibromin in patient samples in results section because the results shows that the mutations in the genes (CDC73 mutations) that encodes the parafibromin is a predictive factor for cervical lymph node metastasis in parathyroid carcinoma.
Thank you for the comment. We have added Figure 1 of representative parafibromin staining images in our manuscript.
Reviewer 3 Report
COMMENTS
The manuscript titled “Patterns and predictors of cervical lymph node metastasis in parathyroid carcinoma” of Ya Hu et al., reports a retrospective analysis of 68 patients affected by parathyroid carcinoma and underwent to cervical lymph node dissection. In 17 of these patients (25.0%) was associated cervical lymph node metastasis. Schulte staging and molecular CDC73 mutations were reported as risk factors for cervical lymph node metastasis. The final proposal is set to include prophylactic cervical lymph node dissection during the initial surgery.
Strength of this investigation can be individualized in the examination of a relatively large cohort of PC. This allows to assess adequate conclusions. Additionally, this paper is well written, and the understanding it expresses is even better for Introduction and Discussion sections.
Minor, Material and Methods as well as Results sections, should be simplified.
Abstract:
Abstract section is adequately describing this study.
Introduction:
Introduction section is adequately describing the aims of study.
Materials and methods:
this section provides sufficient information for the description and reproduction of the study. However, the subdivision of patients should be made easy to understand.
To summarize:
105 patients affected by PC were enrolled for this study.
68 of these patients had available information about a surgical cervical lymph node dissection.
Please, I’d like to ask to author how many patients underwent to cervical lymph node dissection at first PC surgery.
17 patients had metastasis (25%).
Please, I’d like to ask to author how many patients undergone to cervical lymph node dissection at first PC surgery had metastasis.
Primary PC information:
54 patients had available results for parafibromin staining or CDC73 mutations.
Please, I’d like to ask to authors how many of these patients had cervical lymph node metastasis.
52 patients had available results for Ki67 index of primary tumor specimens
Please, I’d like to ask to authors how many of these patients had cervical lymph node metastasis.
Results
The results are adequately described. However,
Line 170: A total of 186 surgeries were performed on this PC patient cohort, including 174 cervical operations.
Should be explain exactly to whom this total refers: to 105 or 68 cohort of patients?
At line 176: Among the 68 patients, 17 (25%) died. Are these patients the same that showed cervical lymph node metastasis?
Tables and Figures:
these give a helpful visual representation of results.
Discussion:
The comments of discussion section are correct and interesting.
Bibliography/References:
References are appropriate.
Decision:
Major, this study has a concrete designation and methods are appropriate to realize the aims.
This study may be accepted for publication after minor revisions.
Author Response
Dear reviewer,
Thank you for your professional and valuable comments. We have revised our manuscript based on these comments, and we hope that these revisions will be met with approval. The changes are highlighted in the revised manuscript, and our point-by-point responses are attached below.
Point 1: Please, I’d like to ask to author how many patients underwent to cervical lymph node dissection at first PC surgery.
Thank you for the comment. 36 patients underwent cervical lymph node dissection during initial surgeries, including 35 patients receiving central LND and one patient receiving lateral LND.
Point 2: Please, I’d like to ask to author how many patients undergone to cervical lymph node dissection at first PC surgery had metastasis.
Thank you for the comment. 7 of 36 patients who underwent cervical LND during initial surgeries had lymph node metastasis.
Point 3: Please, I’d like to ask to authors how many of these patients had cervical lymph node metastasis.
Thank you for the comment. In CDC73 wild-type group, 2 of 25 patients (8%) had cervical lymph node metastasis; however, in CDC73 mutation group, 11 of 29 patients (37.9%) had cervical lymph node metastasis (Figure 3).
Point 4: Please, I’d like to ask to authors how many of these patients had cervical lymph node metastasis.
Thank you for the comment. In the group of Ki-67<5%, 3 of 16 patients (18.8%) had cervical lymph node metastasis; however, in the group of Ki-67≥5%, 10 of 36 patients (27.8%) had cervical lymph node metastasis (Figure 3).
Point 5: Line 170: A total of 186 surgeries were performed on this PC patient cohort, including 174 cervical operations. Should be explain exactly to whom this total refers: to 105 or 68 cohort of patients?
Thank you for the comment. This patient cohort refers to 68 PC patients with lymph node status data. We have revised the manuscript accordingly (Line 190).
Point 6: At line 176: Among the 68 patients, 17 (25%) died. Are these patients the same that showed cervical lymph node metastasis?
Thank you for the comment. These patients were not completely same with those patients who had cervical lymph node metastasis, which was shown in Figure 3.